# Shorter birth intervals between siblings are associated with increased risk of parental divorce

**Venla Berg**[1,2]*, **Anneli Miettinen**[1,3], **Markus Jokela**[4], **Anna Rotkirch**[1]

**1** Population Research Institute, Väestöliitto, Helsinki, Finland, **2** Institute for Molecular Medicine Finland, University of Helsinki, Helsinki, Finland, **3** Kela Research, Kela, Helsinki, Finland, **4** Department of Psychology and Logopedics, University of Helsinki, Helsinki, Finland

* venla.berg@helsinki.fi

**Data Availability Statement:** The FINNUNION data are provided to Population Research Institute by Statistics Finland under specific license for named researchers only and Statistics Finland does not allow it to be shared publicly (Recommendation of

## Abstract

Birth intervals are a crucial component of fertility behaviour and family planning. Short birth intervals are associated—although not necessarily causally—with negative health-related outcomes, but less is known about their associations with family functioning. Here, the associations between birth intervals and marital stability were investigated by Cox regression using a nationally representative, register-based sample of individuals with two (N = 42,481) or three (N = 22,514) children from contemporary Finland (observation period 1972–2009). Shorter interbirth intervals were associated with an increased risk of parental divorce over a ten-year follow-up. Individuals with birth intervals of up to 1.5 years had 24–49 per cent higher divorce risk compared to individuals whose children were born more than 4 years apart. The pattern was similar in all socioeconomic groups and among individuals with earlier and later entry to parenthood. Our results add to the growing body of research showing associations between short birth intervals and negative outcomes in health and family functioning.

## Introduction

The spacing between sibling births in a family is a crucial component of fertility behaviour and family planning. In Western high-income populations, birth intervals seem to have shortened during the late 20[th] and early 21[st] century [1–3]. A large proportion of birth intervals in these populations are now shorter than 2.5 years, which is shorter than what is considered to be the human species-typical birth interval of three to four years, found in hunter–gatherer societies [4]. Women's increased educational levels and labour participation and increased age at first birth in Western countries motivate closer spacing of children [2,5,6]. Further, some family benefits have shown to encourage tight sibling spacing in welfare states [7–10]. Birth intervals have been extensively studied from an epidemiological viewpoint, but less is known about their effects for family functioning. In the current paper we investigate whether sibling spacing is associated with the risk of parental divorce.

the Commission of the European committees on the independence, integrity and accountability of the national and Community statistical authorities (COM(2005) 217 final)). The data can only be used after applying for the permission to use it from Statistics Finland through Population Research Institute. The name of the contact person for this procedure is Tiina Helamaa (tiina.helamaa(at) vaestoliitto.fi).

**Funding:** VB (grant number 266898) and AR (grant numbers 266898 and 260917) were supported by the Academy of Finland (https://www.aka.fi/en/). The funders had no role in study design, data collection and analysis, decision to publish, or preparation of the manuscript.

**Competing interests:** The authors have declared that no competing interests exist.

Very short (less than 18–27 months) and very long (typically over 54–59 months) birth intervals are associated with perinatal health problems for both the mother and the child [11,12]. Long-term associations between shorter intervals and negative health outcomes have also been reported. For example, a few studies have found an association between short birth intervals and reduced longevity in the parents [13–15]. Further, closely spaced siblings have been found to have more behavioural [16] and mental health problems [17–20]. The main mechanism behind the association between short birth intervals and poorer maternal and infant health is hypothesized to be that the mother's body has not had time to fully recover between pregnancies [21]. The risks are most evident in developing countries with limited maternal health care [11,12,21], while contradicting findings have been reported in developed Western countries with advanced health care systems and good nutritional status [22–27]. Importantly, many studies in high-income countries have found that while shorter birth intervals are associated with poorer child health, the association may not be causal [23,24,26,but see 27 for opposite findings].

Much less is known about the associations between birth intervals and sibling relations, parenting, and couple relations. Allegedly, age difference between siblings may affect the psychosocial functioning of family members and family relations, including the marital relationship. Being a parent, especially of young children, is demanding and may affect the wellbeing of parents and the quality of the marital relationship negatively [28,29]. On one hand, closely spaced children have been hypothesized to relief parenting burden because children of same age are assumed to provide each other companionship and because the demands of raising small children are concentrated into a shorter period [see 30 for an early contemplation on the matter]. On the other hand, having closely spaced children may be associated with an increased parenting burden because of the elevated need to provide intensive care simultaneously for many young children [30].

If shorter birth intervals between children were associated with higher parenting stress, we would expect to see an association between shorter intervals and higher likelihood of divorce. Higher parental stress in general [31] and increased parenting stress among parents of children with special needs [32,33] are associated with a higher likelihood of divorce. Giving birth to twins has also been associated with an increased risk of divorce compared to mothers with no twins [34], suggesting that bringing up children close in age may influence divorce risk (although this association may also result from an unexpected increase in the number of children rather than a short birth interval between two children). In addition to infants' need for intensive parental investment, increased parenting stress may be induced by siblings with small age differences having more behavioural [16] and mental health problems [17–20]. Closely spaced siblings also seem to experience an increased risk of serious injury [35] and parental maltreatment [16,36] in childhood, and tend to have more conflictual relationships [37–39].

The evolutionary life history theory provides an additional explanation for the potential association between short birth intervals and higher risk of divorce. According to the theory, individuals vary in their timing, pace, and amount of reproduction in accordance with environmental conditions [40–42]. In resource scarce and dangerous environments with a lower life expectancy, it is more optimal to invest in current reproduction rather than future reproduction because the future is uncertain, and in quantity rather than quality of offspring because of lack of resources [40–44]. These features of faster reproductive strategy lead to shorter birth intervals between children [42]. Faster life history strategy also entails behaviours that destabilize romantic relationships, including decreased parental investment and increased mating effort, i.e., seeking opportunities with new partners [40,43,44]. Empirically, fast life history has been shown to associated with, for example, more negative parenting attitudes, risky

sexual behaviour and higher likelihood of union dissolution [45–47]. Thus, based on life history theory, short birth intervals and increased risk of divorce may correlate even in the absence of a causal, e.g., stress-related, pathway between the two.

Even though absolute poverty is rare in today's high-income societies, relative socioeconomic inequality has been shown to be associated with earlier reproduction, i.e., faster life history, in such societies in many studies. For example, living in poorer neighbourhoods with higher extrinsic mortality, unpredictability, and higher incidence of violent crime is associated with lower age at menarche and lower age at first birth, and more sexual and risky behaviour at young adulthood [e.g., 48–54]. These studies imply that even in prosperous environments, some psychological mechanisms may be sensitive to environmental cues of relative disadvantage and subtly steer our behaviour.

To our knowledge, no previous studies have examined if the length of birth intervals between singleton siblings is associated with parental divorce. Here, we use a nationally representative, register-based dataset of over 60,000 individuals from the late 20th century Finland, and hypothesize that shorter birth intervals of singletons are associated with an increased risk of parental divorce. To take into account potential differences in life history strategies, we examine whether the association is moderated by differences in individuals' adulthood socioeconomic status (SES) and by individuals' age at first reproduction (AFR). We consider lower adulthood SES (at age 35) to be indicative of a faster life history, because lower SES is strongly related to shorter life expectancy and shorter healthy life expectancy in particular [55–57], younger age at first birth [5], and a higher risk of divorce [58]. Earlier entry to parenthood also is, by definition, an indication of faster life history strategy [40]. Thus, among individuals with low SES and younger AFR, short birth intervals and higher divorce risk could correlate even in the absence of any causality between the two. In contrast, we assume people with a high adulthood SES and later entry to parenthood to be on a slower life history path, and an association between short birth intervals and higher divorce risk among them could be indicative of other mechanisms, such as parental stress due to closely spaced children.

## Methods

### Participants

We use register data from Finland, a secularized Nordic welfare state, between years 1972 and 2009. During the study period, the total fertility rate in Finland was between 1.7 and 1.8, and the median number of children among mothers two. The average interbirth interval between first and second births decreased by almost eight months from 3 years 10 months in women born in 1955 to 3 years 2 months in women born in 1975 [2]. In 1987, no-fault divorce legislation was introduced, after which divorce could be initiated without a named reason (e.g., adultery or violence). Consequently, the divorce rate increased considerably and stabilized on a higher level than before 1987. Currently almost 40 per cent of marriages end in divorce in Finland [59]. Further, cohabiting steeply increased during the study period [60], and nowadays it is common to have children out of wedlock in Finland [61]. To a large extent, many cohabiting unions can be seen as socially comparable to marriages. Thus, although this study concentrates on divorces, i.e., dissolving marriages, we decided to include also individuals who got married after the birth of the first child in order to keep the sample as representative as possible.

We use the FINNUNION data set, which is an 11-per cent sample of individuals from cohorts born in 1930–1990 residing in Finland between 1970 and 2010, drawn from the population register by Statistics Finland. The Finnish legislation does not require ethical approval for doing secondary data analysis on existing and anonymized register data [62].

The longitudinal dataset covers complete marital and childbearing histories of the sampled individuals until the end of 2009 (i.e., the data do not contain marital pairs). Finnish register data on divorces is available from January 1972 onwards. We restricted the analyses to cohorts born in 1955 to 1979, whose second children were born on January 1972 earliest. Two separate sets of analyses were performed, one on individuals with two, and only two, children with their first married partner (N = 42,481) and another on individuals with three, and only three, children with their first married partner (N = 22,514). Thus, two-child individuals who continued to have a third child were excluded from the analyses regarding individuals with two children, and three-child individuals who continued to have a fourth child were excluded from the analyses regarding individuals with three children. By thus keeping the study samples as homogenous as possible, we aim to control for some of the confounding exogenous variation, such as the parents of larger families having more traditional values, shorter birth intervals, and lower likelihood of divorce [see, e.g., 63]. Families with twins were excluded from the present study.

First marriages were chosen for the analyses to make the analysis sample as homogenous as possible, because divorce risk is higher on later marriages [58]. Participants who were included in the analyses had to be married before, and still married at, the start of the follow-up (i.e., at the birth of the second or third child, depending on the analysis). The descriptive statistics of the present sample are shown in Table 1.

## Measures

Register data on the birth dates of the participant and participants' children (month and year) were used to calculate participant's *age at first birth* and *interbirth intervals* (IBI; one birth interval in individuals with two children and two birth intervals in individuals with three children). For the analyses, interbirth intervals were categorized into 11 categories, the upper limit of the first one being 18 months, the second 24 months, the third 30 months, etc. The upper limit of the first category, 18 months, was chosen because shorter birth intervals were very rare and if used as a comparison category, would make the statistical analyses non-robust. Date of marriage was used to calculate *marital duration* at the start of follow-up (birth of the second or third child, depending on the analysis). If marriage occurred after the first childbirth, marital duration at the last childbirth was calculated from the birth date of the first child plus six months (assumed minimal time for cohabitation before first childbirth). *Timing of marriage* (1 = Married before first child birth; 2 = Married after first but before second child birth; 3 = Married after second but before third child birth; only applicable in analyses on individuals with three children) was used as an additional covariate to account for the possible differences between unions with and without childbearing out of wedlock. *Quadratic marital duration* was included in all analyses to take into account the intrinsic curvilinear risk of divorce through time within unions [64]. The participant's *birth cohort* (in groups of five years) was included as a covariate to account for the increase in divorce rates during the study period, and participant's *sex* to account for the different proportions of men and women in different socioeconomic groups. The effect of children's sex on birth intervals and parental divorce risk was also examined, but no statistically significant associations between these variables were detected; child sex was thus dropped from the final analyses.

*Participant's SES* at the age of 35 was classified according to Statistics Finland (1 = Farmer; 2 = Entrepreneur/Self-employed; 3 = Upper white-collar; 4 = Lower white-collar; 5 = Manual worker; 6 = Student; 7 = Retired; 8 = Unclassified; 9 = Missing). SES contains information about individuals' main type of activity, occupation, occupational status, and industry and correlates quite strongly with yearly income [65]. SES also correlates with education, but provides

**Table 1. Descriptive statistics.**

| | Women | | Men | |
|---|---|---|---|---|
| | **2 children** | **3 children** | **2 children** | **3 children** |
| N | 22,708 | 11,257 | 19,773 | 9,551 |
| Divorces[a]; n (%) | 3889 (17) | 1441 (13) | 3430 (17) | 1197 (13) |
| Censored persons[a]; n (%) | 447 (2) | 223 (2) | 386 (2) | 172 (2) |
| Average follow-up time in years; mean (SD) | 7.68 (3.14) | 7.77 (3.08) | 7.37 (3.26) | 7.48 (3.21) |
| Birth year;% | | | | |
| 1955–1959 | 30 | 29 | 31 | 32 |
| 1960–1964 | 25 | 27 | 26 | 29 |
| 1965–1969 | 20 | 22 | 20 | 21 |
| 1970–1974 | 14 | 14 | 14 | 13 |
| 1975–1979 | 10 | 8 | 9 | 6 |
| 1st birth interval in years; mean (SD) | 3.29 (1.96) | 2.70 (1.48) | 3.20 (1.85) | 2.62 (1.38) |
| ≤ 18 months; n (%) | 1641 (7) | 1267 (11) | 1453 (7) | 1057 (11) |
| 18.01–24 | 3688 (16) | 2702 (24) | 3372 (17) | 2337 (24) |
| 24.01–30 | 3985 (18) | 2326 (21) | 3593 (18) | 2051 (21) |
| 30.01–36 | 3388 (15) | 1715 (15) | 3019 (15) | 1518 (16) |
| 36.01–42 | 2500 (11) | 1061 (9) | 2207 (11) | 899 (9) |
| 42.01–48 | 1832 (8) | 668 (6) | 1556 (8) | 577 (6) |
| 48.01–54 | 1403 (6) | 452 (4) | 1156 (6) | 352 (4) |
| 54.01–60 | 998 (4) | 278 (2) | 808 (4) | 217 (2) |
| 60.01–66 | 725 (3) | 196 (2) | 604 (3) | 150 (2) |
| 66.01–72 | 464 (2) | 141 (1) | 386 (2) | 86 (1) |
| > 72 | 2084 (9) | 451 (4) | 1619 (8) | 307 (3) |
| 2nd birth interval in years; mean (SD) | | 4.26 (2.48) | | 4.15 (2.37) |
| ≤ 18 months; n (%) | | 534 (5) | | 467 (5) |
| 18.01–24 | | 1213 (11) | | 1035 (11) |
| 24.01–30 | | 1167 (10) | | 1039 (11) |
| 30.01–36 | | 1234 (11) | | 1083 (11) |
| 36.01–42 | | 1023 (9) | | 875 (9) |
| 42.01–48 | | 983 (9) | | 836 (9) |
| 48.01–54 | | 870 (8) | | 714 (7) |
| 54.01–60 | | 772 (7) | | 677 (7) |
| 60.01–66 | | 656 (6) | | 591 (6) |
| 66.01–72 | | 471 (4) | | 429 (4) |
| > 72 | | 2334 (21) | | 1805 (19) |
| Age at first birth; mean (SD) | 26.61 (4.35) | 25.00 (3.77) | 28.52 (4.41) | 27.05 (3.88) |
| Marriage length at last birth in years; mean (SD) | 4.91 (2.56) | 8.27 (3.00) | 4.82 (2.49) | 8.09 (2.88) |
| Timing of marriage; n (%) | | | | |
| Married before first birth | 19045 (84) | 8882 (79) | 16333 (83) | 7470 (78) |
| Married before second birth | 3663 (16) | 1609 (14) | 3439 (17) | 1392 (15) |
| Married before third birth | - | 766 (7) | - | 689 (7) |
| SES at 35; n (%) | | | | |
| Farmer | 848 (2) | 515 (5) | 630 (3) | 575 (6) |
| Entrepreneur /self-employed | 995 (4) | 594 (5) | 1481 (7) | 793 (8) |
| Upper white-collar | 3946 (17) | 1943 (17) | 4289 (22) | 2099 (23) |
| Lower white-collar | 9208 (41) | 4165 (37) | 4018 (19) | 1865 (20) |
| Manual worker | 3439 (15) | 1771 (16) | 6213 (31) | 2950 (33) |

*(Continued)*

**Table 1.** (Continued)

| | Women | | Men | |
|---|---|---|---|---|
| | **2 children** | **3 children** | **2 children** | **3 children** |
| Student | 780 (3) | 440 (4) | 301 (2) | 133 (1) |
| Unclassified | 1787 (8) | 1038 (10) | 1275 (6) | 653 (7) |
| Missing | 2069 (9) | 791 (7) | 1565 (8) | 492 (5) |

[a]During 10-year follow-up. SES = Socioeconomic status; SD = standard deviation.

a more detailed picture of an individual's current social, cultural, and financial position, since some end up in high occupational positions despite not having a university degree, or in low occupational positions despite being highly educated. Information about SES was missing for 3385 and 1706 participants with two children and three children, respectively. About 39 per cent of people with a missing SES had vocational secondary school as their highest degree, and 38 per cent had lower or higher university degree. The individuals with missing information were coded as an additional category and included in the analyses.

## Statistical analyses

The associations between interbirth intervals of children and parental risk of divorce were examined with single-event Cox regression. The dependent variable, i.e., timing of divorces was recorded in months, and follow-up started at the birth of the second child (in analyses on people with two children) or the third child (in analyses on people with three children). The Efron method was used to handle ties [i.e., simultaneous events in the dataset, 66]. The shortest birth interval category (intervals of max. 18 months) was used as the reference category. In models concerning individuals with three children, the first and second birth intervals were examined separately, mutually adjusted, and in interaction, to see if the two intervals had independent, additive, or interactive associations with the risk of divorce. Marital duration and quadratic marital duration at the last childbirth were used as covariates in all models, and thus the hazard ratios can be interpreted as the proportional hazard of a person experiencing a divorce given that the duration of marriage is kept constant. Participants were followed-up until divorce, ten years after the start of follow-up, the end of 2009, participant's emigration or participant's or partner's death, whichever occurred first.

All models adjust for participant's sex, birth cohort, and age at first reproduction, marital duration, quadratic marital duration, and timing of marriage. Additionally, to account for life history strategies, we tested whether or not the associations between birth intervals and risk of divorce differed by SES and AFR by running models with interaction terms between SES and interbirth interval and AFR and interbirth interval.

The proportional-hazards assumption inherent in Cox regression was tested by including an interaction between interbirth interval and analysis time in the Cox model, and on the basis of Schoenfeld residuals [67]. Due to the large sample size, the tests showed significant non-proportionality in the IBI in the analysis of participants with two children, and in the second IBI in the analysis of participants with three children. However, when examining the hazard ratios of different birth interval categories through time, no substantial non-proportionality was detected (i.e., the shortest birth intervals were always associated with the highest divorce risk, followed by the longer birth intervals in ascending order, even though at some points in time the risk ratios were more similar than at some other points). All analyses were carried out with Stata 15.1. [68].

## Results

### Participants with two children

Compared to upper white-collars (estimated marginal mean IBI 39.88 months), farmers and entrepreneurs and self-employed had shorter birth intervals (36.25, 39.22, and 38.27months, respectively). The birth intervals for lower white-collars (40.14 months), manual workers (39.96 months), and students (40.09 months) did not statistically differ from the birth intervals of upper white-collars.

Compared to individuals whose first two children were born at most 18 months apart, individuals whose children were more widely spaced had a lower divorce risk (Table 2, Model 1; for the associations between covariates and risk of divorce, see S1 Table). Adjusting for individual's SES had little effect on the associations between IBI and divorce (Table 2, Model 2). Additional analyses with different IBI categories as the reference group showed that birth intervals up to 3–3.5 years were protective over shorter intervals and longer intervals provided no additional benefit (S2 Table).

To illustrate the associations between the interbirth interval and risk of divorce, we ran a Cox regression model with wider IBI categories (1: ≤18 months; 2: 18.01–24; 3: 24–36; 4: 36–48; 5: > 48), and drew survival functions by IBI categories (Fig 1). Keeping all independent variables at their average, the predicted proportion of individuals who had divorced (vs. were still married) at the end of the follow-up period was 19 per cent (based on the survival function). For individuals with an interbirth interval of no longer than 18 months, the predicted proportion of divorced at the end of follow-up was 26 per cent, and for individuals with an interbirth interval of more than four years, 18 per cent (Fig 1). In other words, individuals with the shortest interbirth intervals were 1.49 times more likely to divorce during the next ten years following the second child's birth than were individuals with the longest interbirth intervals.

**Table 2. Adjusted hazard ratios (HR) from Cox regressions predicting parental risk of divorce by children's interbirth interval in individuals with two children.**

|  | Model 1 |  | Model 2 |  |
| --- | --- | --- | --- | --- |
|  | HR | 95% CI | HR | 95% CI |
| Interbirth interval |  |  |  |  |
| ≤ 18 months | Ref. |  | Ref. |  |
| 18.01–24 | 0.79 | 0.73, 0.87 | 0.80 | 0.73, 0.87 |
| 24.01–30 | 0.69 | 0.63, 0.76 | 0.70 | 0.64, 0.76 |
| 30.01–36 | 0.60 | 0.55, 0.67 | 0.61 | 0.55, 0.67 |
| 36.01–42 | 0.67 | 0.60, 0.75 | 0.68 | 0.61, 0.76 |
| 42.01–48 | 0.62 | 0.55, 0.71 | 0.62 | 0.55, 0.71 |
| 48.01–54 | 0.57 | 0.50, 0.66 | 0.57 | 0.50, 0.66 |
| 54.01–60 | 0.65 | 0.56, 0.77 | 0.65 | 0.56, 0.76 |
| 60.01–66 | 0.66 | 0.55, 0.79 | 0.65 | 0.55, 0.78 |
| 66.01–72 | 0.66 | 0.54, 0.82 | 0.66 | 0.54, 0.82 |
| > 72 | 0.63 | 0.54, 0.75 | 0.63 | 0.53, 0.74 |

All p-values ≤.001.

Model 1 adjusted for birth cohort, sex, age at first reproduction, marriage length at second birth, quadratic marriage length at second birth, and timing of marriage; Model 2 additionally adjusted for SES.

CI = Confidence interval.

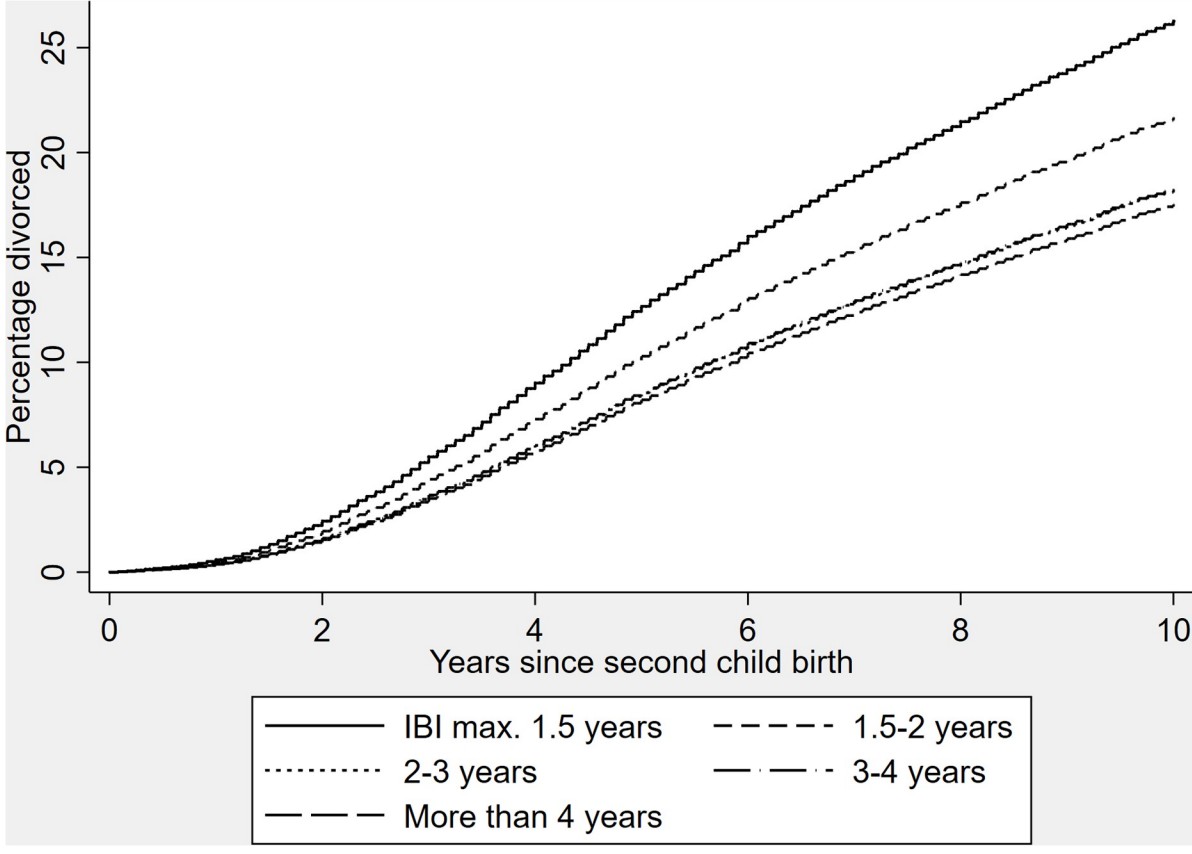

**Fig 1. Cumulative risk of divorce by interbirth interval categories in individuals with two children.** Curves are reversed survival functions from a Cox regression predicting parental risk of divorce, adjusting for parental birth cohort, sex, and age at first birth, marital duration and quadratic marital duration at second birth, timing of marriage, and SES.

### Participants with three children

The effect of the first interbirth interval on the risk of divorce was smaller in three-child families than in two-child families (Table 3), but here also, a first interval of no more than 1.5 years was associated with an increased risk of divorce compared to longer birth intervals (note that Table 3 shows the results from models adjusted for SES). When adjusting for the second birth interval, the effect of the first interval was slightly strengthened (Table 3, Model 3). The length of the second birth interval, however, had no association with individuals' divorce risk.

We further tested whether the second birth interval had differential associations with parental divorce risk depending on the length of the first interval by running a model including both interbirth intervals and their interaction (wider IBI categories along with all the covariates). No significant interactions were found (S3 Table).

Fig 2 illustrates the associations between the two interbirth intervals and risk of divorce, from a Cox regression model with wider IBI categories (adjusting for both interbirth intervals simultaneously as well as for SES). Keeping all independent variables at their average, the predicted proportion of individuals divorced at the end of the follow-up was 14 per cent. For individuals with a first IBI of no longer than 18 months (adjusting for the second IBI), the predicted proportion of divorced at the end of follow-up was 17 per cent, and for individuals with a first IBI of more than four years, 14 per cent (Fig 2a). In other words, the marriages of individuals with a short first IBI were 1.24 times more likely to dissolve than were the marriages of

**Table 3. Hazard ratios (HR) from Cox regressions predicting parental risk of divorce by interbirth intervals (IBI) in individuals with three children.**

| | Model 1 | | | Model 2 | | | Model 3 | | |
|---|---|---|---|---|---|---|---|---|---|
| | HR | 95% CI | p | HR | 95% CI | p | HR | 95% CI | p |
| 1st IBI | | | | | | | | | |
| ≤ 18 months | Ref. | | | | | | Ref. | | |
| 18.01–24 | 0.79 | 0.70, 0.90 | .000 | | | | 0.79 | 0.69, 0.89 | .000 |
| 24.01–30 | 0.74 | 0.65, 0.85 | .000 | | | | 0.73 | 0.64, 0.84 | .000 |
| 30.01–36 | 0.70 | 0.60, 0.81 | .000 | | | | 0.68 | 0.59, 0.80 | .000 |
| 36.01–42 | 0.85 | 0.72, 1.00 | .052 | | | | 0.83 | 0.70, 0.99 | .042 |
| 42.01–48 | 0.78 | 0.64, 0.94 | .010 | | | | 0.76 | 0.61, 0.93 | .008 |
| 48.01–54 | 0.89 | 0.72, 1.11 | .313 | | | | 0.87 | 0.69, 1.10 | .243 |
| 54.01–60 | 0.79 | 0.60, 1.03 | .087 | | | | 0.76 | 0.57, 1.01 | .063 |
| 60.01–66 | 0.72 | 0.52, 0.99 | .042 | | | | 0.69 | 0.49, 0.97 | .032 |
| 66.01–72 | 0.84 | 0.58, 1.21 | .343 | | | | 0.80 | 0.54, 1.18 | .267 |
| > 72 | 0.85 | 0.68, 1.07 | .167 | | | | 0.82 | 0.61, 1.10 | .183 |
| 2nd IBI | | | | | | | | | |
| ≤ 18 months | | | | Ref. | | | Ref. | | |
| 18.01–24 | | | | 1.10 | 0.91, 1.33 | .304 | 1.10 | 0.91, 1.33 | .336 |
| 24.01–30 | | | | 1.12 | 0.92, 1.35 | .266 | 1.10 | 0.90, 1.34 | .347 |
| 30.01–36 | | | | 1.05 | 0.86, 1.28 | .633 | 1.02 | 0.83, 1.25 | .839 |
| 36.01–42 | | | | 1.00 | 0.82, 1.24 | .964 | 0.97 | 0.78, 1.21 | .798 |
| 42.01–48 | | | | 0.98 | 0.79, 1.22 | .889 | 0.94 | 0.74, 1.18 | .586 |
| 48.01–54 | | | | 1.07 | 0.85, 1.33 | .583 | 1.02 | 0.80, 1.30 | .887 |
| 54.01–60 | | | | 0.85 | 0.67, 1.09 | .207 | 0.80 | 0.61, 1.05 | .115 |
| 60.01–66 | | | | 1.05 | 0.82, 1.35 | .680 | 0.98 | 0.75, 1.30 | .908 |
| 66.01–72 | | | | 0.94 | 0.71, 1.24 | .658 | 0.87 | 0.64, 1.19 | .386 |
| > 72 | | | | 1.15 | 0.92, 1.45 | .216 | 1.06 | 0.80, 1.41 | .684 |

All models are adjusted for birth cohort, sex, age at first reproduction, marital length at 3rd birth, quadratic marital length at 3rd birth, timing of marriage, and socio-economic status. Model 1: First IBI only; Model 2: 2nd IBI only; Model 3: Both IBIs. CI = Confidence interval.

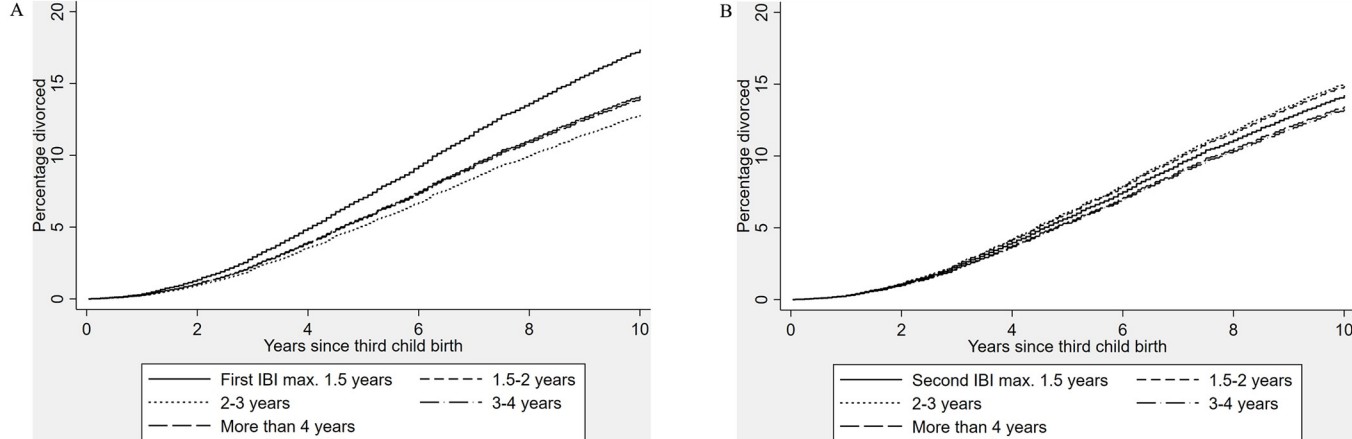

**Fig 2. Cumulative risk of divorce by the first (a) and second (b) interbirth interval in individuals with three children.** Curves are reversed survival functions of a Cox regression predicting parental risk of divorce, controlling for parental birth cohort, sex, and age at first birth, marital duration and quadratic marital duration at third birth, timing of marriage, SES, and for both interbirth intervals simultaneously.

individuals with a long first IBI in the ten-year follow-up. The second birth interval was not associated with divorce risk (Fig 2b) among individuals with three children.

## Differences in associations by SES and AFR

We then examined the interactions between SES and birth intervals and AFR and birth intervals to investigate the role of life history strategy in the association between birth intervals and divorce. In order to avoid numerous interactions between multiple birth interval and socioeconomic or AFR categories, we divided birth intervals into two classes (1.5 years and under, and over 1.5 years) for these analyses. None of the interactions among individuals with two and three children between SES and the length of interbirth intervals were significant (S4 Table), indicating that sibling spacing was similarly associated with divorce risk in all SES groups. Also the interactions between AFR and IBI among individuals with two and three children were mostly non-significant, indicating the sibling spacing was similarly associated with the risk of divorce regardless of the age of entry to parenthood (S5 Table). The only exception here was the interaction between IBI and entering parenthood at age 30–34, which was just significant (HR = 0.80, 95% CI = 0.65–1.00, p = .049), indicating that in this AFR group, longer IBI (more than 18 months compared to max. 18 months) was more strongly associated with lower risk of divorce compared to individuals who entered parenthood before turning 25 (HR = 0.62, 0.51–0.76 and HR = 0.78, 0.70–0.87, respectively). This interaction effect is in the opposite direction than what we hypothesized based on life history theory (weaker association in individuals with a slower life history strategy, i.e., later start of childbearing).

## Sensitivity analyses

To see whether the arbitrary choice of relationship length for individuals who got married after the birth of the first child affected our results, we ran analyses where only the true length of the actual marriage was taken into account (regardless of the timing of marriage), and including only individuals who were married before the birth of the first child. The results were very similar to the main analyses (Fig 3).To see if our choice of study sample affected the results, we ran sensitivity analyses including all individuals regardless of their final number of children (censoring at the birth of the third child in the model where follow-up started at the second child, and at the birth of the fourth child in the model where follow-up started at the third child). In these analyses, the association between longer intervals and lower risk of divorce persisted, but was attenuated (Fig 3 and S6 Table). Note that this analysis strongly confounds higher number of children, lower risk of divorce, and shorter birth intervals—because individuals with more than two children with the same person very rarely divorce before the birth of the third child, and have substantially shorter first birth intervals (Table 1).

## Discussion

Short birth intervals between siblings have previously been shown to be associated (although not necessarily causally) with a range of negative perinatal and long-term health outcomes for parents and children [11,12] and more conflictual sibling relations [37,38]. Here, we showed in a large, nationally representative sample of Finns, that individuals whose children were more closely spaced had an increased risk of divorce, as hypothesized. In a ten-year follow-up, individuals with two children born no more than 1.5 years apart had a 49 per cent higher divorce risk compared to individuals with two children born more than 4 years apart. Consistent with previous research concerning the associations between number of children and parental divorce risk [58,64], individuals with three children were somewhat less likely to divorce than those with two children. But also among individuals with three children, a short

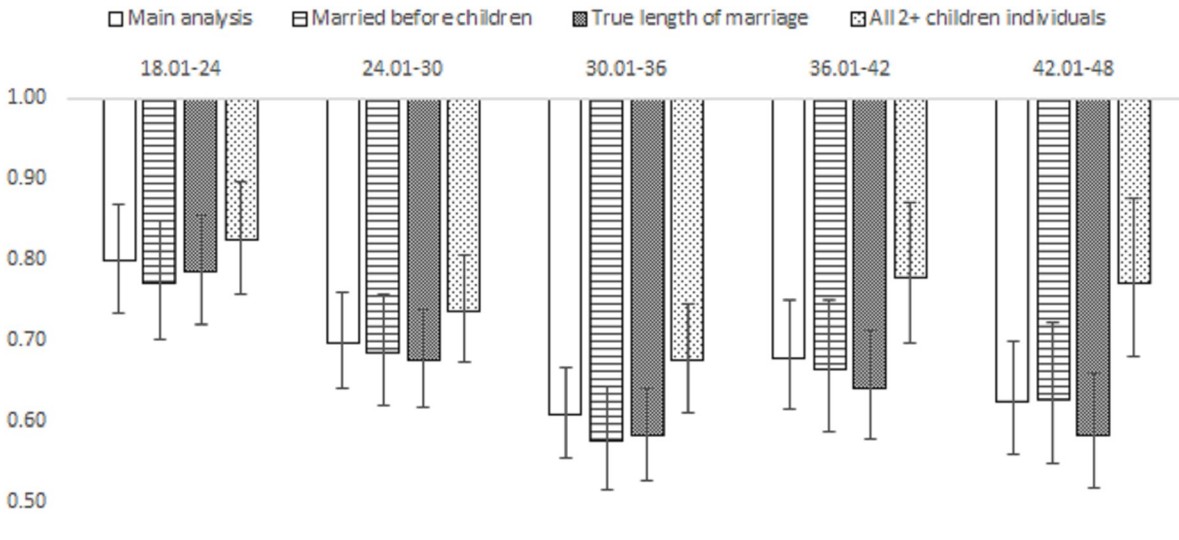

**Fig 3. Hazard ratios (HR) from Cox regressions predicting the risk of divorce by the first IBI.** Reference category is max. 18 months, and showing HRs for the five next IBI categories. Results from main analyses (Table 2, Model 2, controlling for SES) and from different sensitivity analyses. Married before children: Model includes only individuals who were married at the birth of the first child. True length of marriage: Model controls for the length of marriage only, regardless of the timing of marriage (even if married after the birth of first child). All 2+ children individuals: Includes everyone who had at least two children.

birth interval between the first and second child was associated with an increased risk of parental divorce. Regardless of the length of the second birth interval, individuals with a first birth interval of no more than 1.5 years had 24 per cent higher divorce risk compared to individuals whose first birth interval was more than 4 years. The associations found in this study were little affected by adjusting for SES. Further, the associations were equally strong in all socioeconomic groups and among individuals who entered parenthood earlier and later.

We assumed that the higher divorce risk among parents of closely spaced children could stem from elevated daily stress between family members. We also assumed that a fast life history strategy could induce a correlation between the two among people with a lower SES or earlier entry to parenthood, irrespective of elevated daily stress. These mechanisms are, naturally, not mutually exclusive. We hypothesized that in the absence of a causal mechanism related to, for example, stress, the association between shorter intervals and higher divorce risk could still be present among individuals with a low SES or young AFR, but absent or less strong among the high SES or individuals entering parenthood later. The interaction analyses showed that shorter birth intervals were similarly correlated with a higher divorce risk in low as well as high socioeconomic groups and in those who started childbearing earlier vs. later. In other words, even among people who can be hypothesized to follow a slow life history strategy (i.e., people in high adulthood socioeconomic position or later entry to parenthood), closely spaced children were associated with an increased risk of divorce. Higher parental stress is one plausible pathway between the two phenomena.

However, in individuals with three children, the first birth interval was associated with divorce risk while the second birth interval was not (see Table 3 and Fig 2). On one hand, the alleged stress posed by first two closely spaced children appears to persist irrespective of the timing of the third child. On the other hand, a third child born within a short time period does not seem to increase the alleged parental stress—at least not in ways that would be detrimental

to the marital union. It might be that the profound life changes associated with becoming a parent both for the individuals [69–71] and the couple relationship [72] present a particularly vulnerable period during which the marital union can be permanently shaken by a rapid birth of the next child. Above and beyond this risk the timing of the third birth seems to make no difference.

Given that our study is observational, one must, however, abstain from making causal inferences based on the association between children's birth intervals and parental risk of divorce. Even though we controlled for several potential confounders and checked for potential moderating effects of life history strategy, it is possible that short birth intervals and divorce risk reflect the consequences of yet some other variables that were not included in our models. For example, parents with marital problems might attempt to overcome those problems by having more children faster. Further, some genetically varying tendencies such as impulsivity might be associated with both shorter pregnancy intervals and the likelihood of divorce [73]. Quasi-experimental studies based on exogenous variation in birth intervals would be needed to investigate the causality in the association between birth intervals and parental risk of divorce further.

One of the main strengths of the current study is the large, nationally representative dataset with highly reliable register data on dates of births, marriages, and divorces, spanning a period of 37 years. Among the limitations is the fact that the data may underestimate the proportion of children born to men, because not all biological fathers are registered. During the study period, however, the proportion of Finnish children without a registered father was only around 2 per cent [74]. A large majority of these children of non-registered fathers would have been born outside marriages—children born in marriages (around 80 per cent of the present study sample) are automatically registered to both members of the marital union. It is thus safe to say that the possible underestimation of children born to men has not severely affected the present analyses. A second caveat is that the data only had reliable data on divorces, while cohabitation was very common in Finland during the study period. Future studies should employ data with reliable information on all types of union dissolution.

The results of our study add to the growing body of evidence of negative outcomes associated with short birth intervals [75]. Nevertheless, with the rising age at first birth evident in many Western countries, short interbirth intervals are likely to become more common. Many welfare states, Finland including, also implement family benefits that encourage shorter intervals. The implementation of these types of family benefits seem to have decreased the average interbirth interval in Sweden, Australia, and Finland, above and beyond the effect of age at first birth [2,7–10]. The possible ramifications of short birth intervals may be further intensified by the limited social support from kin and peers that characterizes modern Western societies [76]. It has been suggested, and in some Western countries already implemented, that family planning services should expand their focus from how to *prevent* unwanted pregnancies to how to *achieve* the desired number of children. This includes, for example, advice on the optimal timing of wanted pregnancies [see, e.g., 77]. The present findings, alongside previous study results on short birth intervals, suggest that there is an increasing demand of such family planning knowledge in the broad sense, aimed at both policy makers and the general public.

## Supporting information

**S1 Table. Hazard ratios from Cox regressions predicting parental risk of divorce by all the covariates (mutually adjusted) in individuals with two and three children.**
(PDF)

**S2 Table. Results from Cox regressions predicting the risk of divorce by interbirth interval (IBI) between children in two-child families, with varying reference categories.**
(PDF)

**S3 Table. Hazard ratios (HR) from Cox regression predicting risk of divorce in individuals with three children by first and second birth interval and their interaction.**
(PDF)

**S4 Table. Hazard ratios from Cox regressions predicting the risk of divorce by interbirth (IBI) intervals, socioeconomic status (SES), and their interactions.**
(PDF)

**S5 Table. Hazard ratios from Cox regressions predicting the risk of divorce by interbirth (IBI) intervals, age at first reproduction (AFR), and their interactions.**
(PDF)

**S6 Table. Results from Cox regressions predicting the risk of divorce by birth intervals in all individuals regardless of their final number of children.**
(PDF)

## Acknowledgments

The data were provided by Statistics Finland.

## Author Contributions

**Conceptualization:** Venla Berg, Anna Rotkirch.

**Data curation:** Anneli Miettinen.

**Formal analysis:** Venla Berg.

**Resources:** Anna Rotkirch.

**Writing – original draft:** Venla Berg.

**Writing – review & editing:** Venla Berg, Anneli Miettinen, Markus Jokela, Anna Rotkirch.

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
