## [Decision Letter · Decision Letter 0]

18 Nov 2019

PONE-D-19-28900

Shorter Birth Intervals between Siblings Are Associated with Increased Risk of Parental Divorce

PLOS ONE

Dear Dr Berg,

Thank you for submitting your manuscript to PLOS ONE. After careful consideration, we feel that it has merit but does not fully meet PLOS ONE’s publication criteria as it currently stands. Therefore, we invite you to submit a revised version of the manuscript that addresses the points raised during the review process.

We would appreciate receiving your revised manuscript by Dec 27 2019 11:59PM. To enhance the reproducibility of your results, we recommend that if applicable you deposit your laboratory protocols in protocols.io, where a protocol can be assigned its own identifier (DOI) such that it can be cited independently in the future. For instructions see: http://journals.plos.org/plosone/s/submission-guidelines#loc-laboratory-protocols

We look forward to receiving your revised manuscript.

Kind regards,

David Meyre

Academic Editor

PLOS ONE

Journal Requirements:

Reviewers' comments:

Reviewer's Responses to Questions

**Comments to the Author**

1. Is the manuscript technically sound, and do the data support the conclusions?

Reviewer #1: Partly

Reviewer #2: Yes

2. Has the statistical analysis been performed appropriately and rigorously? 

Reviewer #1: Yes

Reviewer #2: Yes

3. Have the authors made all data underlying the findings in their manuscript fully available?

Reviewer #1: No

Reviewer #2: No

4. Is the manuscript presented in an intelligible fashion and written in standard English?

Reviewer #1: Yes

Reviewer #2: Yes

5. Review Comments to the Author

Reviewer #1: PLOS One review

Summary: The current manuscript uses Finnish population data to examine the relation between interbirth intervals and divorce rates over a 37 year span of time with an average follow-up of 10 years. They find that shorter intervals are associated with increased divorce rates. The paper would be improved by a reorganization of the introduction and a greater consideration of alternative explanations.

Abstract:

1. Adjust language that it is established that short interbirth intervals are associated with adverse perinatal outcomes, as this is still somewhat debated.

Introduction:

1. The first paragraph of the introduction briefly provides some history of interpregnancy interval evolutionary theories, but would benefit from a more unbiased initial presentation of the studies and evidence. First, there have been some challenges to the assumption that shorter birth intervals are causally associated with adverse birth outcomes (see Ball et al., 2014, Class et al., 2017 for example). The WHO recommendations are outdated and based on 1 study that did not do an adequate job of controlling for confounding factors. Second, one can alternatively imagine that increased maternal education levels and workforce participation has contributed to delayed maternal childbearing and an elongation of interpregnancy intervals. Third, work has also shown that long interpregnancy intervals are similarly problematic – also associated with adverse birth outcomes (like a ‘U’ shaped curve). I think this issues can simply be addressed by including some of these alternate perspectives and citations.

As I continued to read the introduction, the above mentioned topics were discussed. Therefore, perhaps a reordering of the presentation may help to paint a less one-sided take to the evidence.

2. This claim is uncited: “Raising two or more children of similar age is even more demanding, and the elevated need to provide intensive care is likely to affect parental stress” and is an assumption. The citation that occurs a few sentences later is in regards to twins and does not support this claim. This claim needs citation as, it could alternatively be the case that having two children of similar age is beneficial for the parents as the parents are therefore more familiar with the developmental needs and abilities of a certain age group, they can participate in similar activities with their children rather than having to meet the needs of two different levels all the time, even daily hassles such as similar bedtimes, napping needs, and feeding abilities would be lessened. This is all to say that arguments can be made in either direction, but if they are made, they need to be cited or framed as hypotheses. This again, may be addressed by rearranging the presentation of the introduction.

Methods:

1. Can you explain the separation of Women and Men in Table 1? Is it the case that these might be the same children? Am I correct in understanding that the parental sex was included in models, but not the child’s sex?

2. It would be helpful to have percentages for all variables (e.g. divorces) in Table 1.

3. Was it possible to use interpregnancy interval as opposed to interbirth interval (thus accounting for short pregnancies influencing the possibility of shorter birth intervals)?

4. Please make more clear what your reference intervals are. As it is stated in the abstract, one might imagine that interbirth intervals over 4 years are the reference.

5. What is the reasoning behind the choice to look at the first birth interval and the second birth interval separately within the 3 child families? Why would one believe that those two intervals contribute to divorce differently? It seems that this possibility is not introduced until the discussion and renders investigating if there is a difference between those intervals as slightly unfounded.

6. Using different interval spacing models makes your findings a bit difficult to synthesize. I believe your findings would translate more cleanly to previous research if a consistent model was used throughout the paper and those categories were similar to what previous work has used (perhaps being interbirth rather than interpregnancy).

Discussion:

1. This reviewer appreciates the limitations of causal interpretations the authors provide. These explanations might deserve more attention as they appear stronger than the explanation of stress caused by having two kids close together causing divorce. The authors state that quasi-experimental studies are needed to “strengthen” the argument of a causal relation between short interbirth interval and divorce, but I don’t believe that this study has the ability to make that initial claim at all. Perhaps a quasi-experimental study is needed to challenge the explanation that people who have closely spaced children also have a higher likelihood of divorce (correlation, but not causal relation).

Minor:

1. Gender is used interchangeably with sex and should not be.

2. The abbreviation of SES is not defined when first used and then inconsistently used.

3. Line 330: Obstain not obtain.

Reviewer #2: This is in many ways a competent and interesting manuscript. However, I have concerns about how the research addresses the underlying hypothesis, which is that short birth intervals negatively affect family functioning and in turn increase subsequent divorce, in part due to maternal depletion: the analysis does not test this pathway. Second, I question the use of socioeconomic status as a proxy variable representing fast life-history strategies. The life-history strategy argument is not made clearly enough as a theoretical approach: the authors need to convince the reader that short birth intervals are part of a fast life-history strategy. Usually it is only applied to the timing of maturity and reproduction, not to short birth intervals. Under this hypothesis, low socioeconomic status couples are presumably focussed on offspring quantity over quality, and short birth intervals with subsequent divorce and remarriage could be part of this reproductive strategy.

It would be worth developing and tightening the theoretical framework and determining whether the FINNUNION data can more directly address any hypothesised mechanisms linking short birth intervals and the likelihood of divorce. This is not easy, as FINNUNION is specifically focussed on marriage and does not appear to contain relevant intermediary variables. One possibility is related to the idea that boys increase maternal depletion relative to girls. For example, women consume more calories during pregnancy with a male offspring (Tamimi et al. 2003). If depletion is a root cause of marital stress and subsequent divorce, then a boy-boy pair with short birth spacing should be the most depleting for mothers, and a girl-girl pair should be the least depleting. Perhaps instead of dropping infant sex from the analyses you could use the infant sex data to more directly address the maternal depletion-related hypothesis.

An alternative hypothesis that could be tested with the FINNUNION sample is that rather than being related to fast versus slow life-histories, short birth intervals followed by divorce could result from alternative male mating strategies. If, like in other nations, divorce results in an increased maternal childcare burden and reduced paternal involvement (Kalmijn 2015), short birth intervals followed by divorce could be a male mating strategy: have a few children closely spaced then move on to a new partner. This can be addressed using the Finnish data: controlling for age, do men who divorce after short birth intervals marry again sooner than men who divorce after longer birth intervals? This would entail an analysis of divorced individuals, with birth intervals predicting time from divorce to remarriage

Minor comments:

Summarize sensitivity analyses in the main part of the paper: readers will want to see what difference it makes to use particular subsamples and different model specifications. In STATA, there is an add-on for creating coefficient plots to graphically summarise hazard ratios from several separate statistical analyses (Jann 2014).

I don’t understand why birth interval was categorised – why not keep it as a continuous independent variable?

The gap of divorce by years since 2nd birth continues to widen between birth interval groups long beyond the period of high infant dependency. Is it still stressful to have a short birth interval when they are both in school?

References:

Tamimi et al, (2003). Average energy intake among pregnant women carrying a boy compared with a girl. BMJ 326:1245-6.

Kalmijn, M. (2015). Father-Child Relations after Divorce in Four European Countries: Patterns and Determinants. Comparative Population Studies, 40, n. 3.

Jann B. (2014). Plotting regression coefficients and other estimates. The Stata Journal 2011;14(4):708-37.

6. PLOS authors have the option to publish the peer review history of their article (what does this mean?). If published, this will include your full peer review and any attached files.

Reviewer #1: No

Reviewer #2: Yes: David Waynforth

---

## [Author Response · Author response to Decision Letter 0]

26 Dec 2019

Response to reviewers.

We thank the anonymous reviewer and Dr. David Waynforth for the time and effort they have invested in our study, and for the insightful comments and critique they have provided. Below are our point by point answers to their comments.

Reviewer #1: PLOS One review

Summary: The current manuscript uses Finnish population data to examine the relation between interbirth intervals and divorce rates over a 37 year span of time with an average follow-up of 10 years. They find that shorter intervals are associated with increased divorce rates. The paper would be improved by a reorganization of the introduction and a greater consideration of alternative explanations.

Abstract:

1. Adjust language that it is established that short interbirth intervals are associated with adverse perinatal outcomes, as this is still somewhat debated.

Author response #1: We have adjusted the language in the abstract.

Introduction:

1. The first paragraph of the introduction briefly provides some history of interpregnancy interval evolutionary theories, but would benefit from a more unbiased initial presentation of the studies and evidence. First, there have been some challenges to the assumption that shorter birth intervals are causally associated with adverse birth outcomes (see Ball et al., 2014, Class et al., 2017 for example). The WHO recommendations are outdated and based on 1 study that did not do an adequate job of controlling for confounding factors. Second, one can alternatively imagine that increased maternal education levels and workforce participation has contributed to delayed maternal childbearing and an elongation of interpregnancy intervals. Third, work has also shown that long interpregnancy intervals are similarly problematic – also associated with adverse birth outcomes (like a ‘U’ shaped curve). I think this issues can simply be addressed by including some of these alternate perspectives and citations.

As I continued to read the introduction, the above mentioned topics were discussed. Therefore, perhaps a reordering of the presentation may help to paint a less one-sided take to the evidence.

Author response #2: We have revised the beginning of the introduction in many ways: i) We deleted the reference to the WHO guideline; ii) we deleted the part referring to short intervals being risky from the first paragraph; iii) we made more clear that the idea that postponement of parenthood and higher education among women are associated with shorter birth intervals is based on research findings, not an assumption; iv) we mention the U-shaped association between birth intervals and health outcomes right at the beginning or the paragraph discussing these findings; and v) we have more explicitly stated that the causality between birth intervals and health outcomes is currently under debate.

2. This claim is uncited: “Raising two or more children of similar age is even more demanding, and the elevated need to provide intensive care is likely to affect parental stress” and is an assumption. The citation that occurs a few sentences later is in regards to twins and does not support this claim. This claim needs citation as, it could alternatively be the case that having two children of similar age is beneficial for the parents as the parents are therefore more familiar with the developmental needs and abilities of a certain age group, they can participate in similar activities with their children rather than having to meet the needs of two different levels all the time, even daily hassles such as similar bedtimes, napping needs, and feeding abilities would be lessened. This is all to say that arguments can be made in either direction, but if they are made, they need to be cited or framed as hypotheses. This again, may be addressed by rearranging the presentation of the introduction.

Author response #3: We have revised this part of the introduction to include the potential positive outcomes of tighter sibling spacing, and more clearly state the speculative / hypothetical nature of our assumption on short birth intervals and increased parental stress.

Methods:

1. Can you explain the separation of Women and Men in Table 1? Is it the case that these might be the same children? Am I correct in understanding that the parental sex was included in models, but not the child’s sex?

Author response #4: The data include individuals, not married couples, but since this is a random 11% sample of Finns, it is possible that a small fraction of the sample are individuals married to each other (i.e., same children). Yes, parental sex was included in the model, because it is known that men are more likely to divorce (a larger proportion of men than women never marry, and more men have multiple marriages than women). To control for this in a sample including both men and women, we included parental sex in the models. Children’s sex was not associated with parental divorce risk, so it was left out from the final models.

2. It would be helpful to have percentages for all variables (e.g. divorces) in Table 1.

Author response #5: We added the percentages in Table1, wherever they were lacking.

3. Was it possible to use interpregnancy interval as opposed to interbirth interval (thus accounting for short pregnancies influencing the possibility of shorter birth intervals)?

Author response #6: Unfortunately, our data do not contain information on gestational length, so we could only examine birth intervals, not interpregnancy intervals. 

4. Please make more clear what your reference intervals are. As it is stated in the abstract, one might imagine that interbirth intervals over 4 years are the reference.

Author response #7: We have added a mention on the reference interval in the “Statistical analysis”-part of the Methods-section, and also state it now explicitly in all the tables. In the abstract, in our opinion, it would still be best to keep the comparison as it is, as XX% higher risk is more intuitive and easy to understand than XX% lower risk (naturally, this comparison would have stayed exactly the same even if we had had the longest interval as a reference category).

5. What is the reasoning behind the choice to look at the first birth interval and the second birth interval separately within the 3 child families? Why would one believe that those two intervals contribute to divorce differently? It seems that this possibility is not introduced until the discussion and renders investigating if there is a difference between those intervals as slightly unfounded.

Author response #8: To us, this was an intuitive choice because 3-child families have two birth intervals. We did not have a specific hypothesis about this, but feel looking at them separately is the most open way of examining the effects of the two intervals. We have now justified our choice explicitly: “In models concerning individuals with three children, the first and second birth intervals were examined separately, mutually adjusted, and in interaction, to see if the two intervals had independent, additive, or interactive associations with the risk of divorce.”

6. Using different interval spacing models makes your findings a bit difficult to synthesize. I believe your findings would translate more cleanly to previous research if a consistent model was used throughout the paper and those categories were similar to what previous work has used (perhaps being interbirth rather than interpregnancy).

Author response #9: The paper uses interbirth intervals as the predictor (not interpregnancy intervals), and we do not have information on interpregnancy. In studies with interpregnancy interval as the main predictor, the shortest category is usually max. 5 months (corresponding roughly to an interbirth interval of 14 months) – however, our shortest reference category is a bit longer (18 months) because we didn’t have enough cases in the shorter categories to maintain statistical power. Different limits for the highest birth interval category in different analyses were used because we wanted to keep as many separate categories for the longer intervals as possible to investigate the possibility of a U-shaped effect between birth intervals and divorce risk. Very long intervals were so rare in the first IBI for people with 3 children that the categorization here was different. However, we have now changed the categorization in all analyses to be similar, that is, highest category is now always longer than 72 months.

Discussion:

1. This reviewer appreciates the limitations of causal interpretations the authors provide. These explanations might deserve more attention as they appear stronger than the explanation of stress caused by having two kids close together causing divorce. The authors state that quasi-experimental studies are needed to “strengthen” the argument of a causal relation between short interbirth interval and divorce, but I don’t believe that this study has the ability to make that initial claim at all. Perhaps a quasi-experimental study is needed to challenge the explanation that people who have closely spaced children also have a higher likelihood of divorce (correlation, but not causal relation).

Author response #10: We have revised this bit of text, and it now reads: ”Quasi-experimental studies based on exogenous variation in birth intervals would be needed to investigate the causality in the association between birth intervals and parental risk of divorce.”

Minor:

1. Gender is used interchangeably with sex and should not be.

2. The abbreviation of SES is not defined when first used and then inconsistently used.

3. Line 330: Obstain not obtain.

Author response #11: Thank you, these mistakes are now corrected.

Reviewer #2: This is in many ways a competent and interesting manuscript. However, I have concerns about how the research addresses the underlying hypothesis, which is that short birth intervals negatively affect family functioning and in turn increase subsequent divorce, in part due to maternal depletion: the analysis does not test this pathway. 

Author response #12: Please see Author response #14. 

Second, I question the use of socioeconomic status as a proxy variable representing fast life-history strategies. The life-history strategy argument is not made clearly enough as a theoretical approach: the authors need to convince the reader that short birth intervals are part of a fast life-history strategy. Usually it is only applied to the timing of maturity and reproduction, not to short birth intervals. Under this hypothesis, low socioeconomic status couples are presumably focussed on offspring quantity over quality, and short birth intervals with subsequent divorce and remarriage could be part of this reproductive strategy.

Author response #13: We have made the point of short birth intervals being a part of fast life history more clear in the introduction: “In resource scarce and dangerous environments with a lower life expectancy, it is more optimal to invest in current reproduction rather than future reproduction because the future is uncertain, and in quantity rather than quality of offspring because of lack of resources (40–44). These features of faster reproductive strategy lead to shorter birth intervals between children (42).”

To accommodate the reviewers concern about SES being a good indicator of fast life history strategy, we conducted a new analysis examining the interaction between age at first reproduction and length of birth intervals with a similar assumption than the SES-interaction analyses (older age at entry to parenthood is indicative of slower life history and association between IBI and divorce should be weaker/absent among these people if life history alone would explain the association). The results for this additional analysis are in accordance with and confirm the SES-interaction analyses: no significant interactions were found (except for one just significant in the opposite direction than hypothesized). These new results are given in supplementary table S5, and summarized in the text.

It would be worth developing and tightening the theoretical framework and determining whether the FINNUNION data can more directly address any hypothesised mechanisms linking short birth intervals and the likelihood of divorce. This is not easy, as FINNUNION is specifically focussed on marriage and does not appear to contain relevant intermediary variables. One possibility is related to the idea that boys increase maternal depletion relative to girls. For example, women consume more calories during pregnancy with a male offspring (Tamimi et al. 2003). If depletion is a root cause of marital stress and subsequent divorce, then a boy-boy pair with short birth spacing should be the most depleting for mothers, and a girl-girl pair should be the least depleting. Perhaps instead of dropping infant sex from the analyses you could use the infant sex data to more directly address the maternal depletion-related hypothesis.

Author response #14: With this comment, we realize that we have been somewhat unsuccessful in explaining the assumed pathway from short birth intervals to parental stress and divorce. We do not assume that this pathway would result from maternal depletion during pregnancy, but rather the strains of everyday life related to bringing up children of similar age – affecting mothers as well as fathers. We have altered the introduction to make our point clearer and to state more openly the hypothetical nature of this stress-related pathway (see Author response # 3). For this reason, we think that testing the maternal depletion hypothesis the way suggested by the reviewer would not be compatible with the research questions in this study, and have not conducted them (also, the sexes of children were not associated with divorce risk when we tested them in initial analyses). 

An alternative hypothesis that could be tested with the FINNUNION sample is that rather than being related to fast versus slow life-histories, short birth intervals followed by divorce could result from alternative male mating strategies. If, like in other nations, divorce results in an increased maternal childcare burden and reduced paternal involvement (Kalmijn 2015), short birth intervals followed by divorce could be a male mating strategy: have a few children closely spaced then move on to a new partner. This can be addressed using the Finnish data: controlling for age, do men who divorce after short birth intervals marry again sooner than men who divorce after longer birth intervals? This would entail an analysis of divorced individuals, with birth intervals predicting time from divorce to remarriage

Author response #15: This is an excellent idea, and definitely worth investigating – however, we feel that it would be a topic of a different study (we would be happy collaborate with the reviewer in conducting such a study). In this paper, we are mainly interested in establishing an association between birth intervals and risk of divorce, which is a new finding in itself, and be cautious about this association not necessarily being causal. In our opinion, adding analyses with follow-up starting at divorce would draw the reader’s attention away from the main point of the current study. 

Minor comments:

Summarize sensitivity analyses in the main part of the paper: readers will want to see what difference it makes to use particular subsamples and different model specifications. In STATA, there is an add-on for creating coefficient plots to graphically summarise hazard ratios from several separate statistical analyses (Jann 2014).

Author response #16: We plotted the HRs from main analyses and sensitivity analyses in what is now Fig. 3, and added a section called “Sensitivity analyses” at the end of Results, summarizing the analyses.

I don’t understand why birth interval was categorised – why not keep it as a continuous independent variable?

Author response #17: Birth intervals are regularly categorized in literature examining this topic, and in order to keep our results comparable with the literature, we have categorized the birth intervals (see also Author response #9).

The gap of divorce by years since 2nd birth continues to widen between birth interval groups long beyond the period of high infant dependency. Is it still stressful to have a short birth interval when they are both in school?

Author response #18: Divorce processes in marital relationships are usually long and take years to develop. Stress related to bringing up closely spaced siblings in infancy may have long-term ramifications on the marital relationship that only manifest later (see also MS lines 351-354).

Author note: In addition to the changes requested by the reviewers, we have made a small number of changes to improve the clarity of the manuscript.

---

## [Editor Report · Decision Letter 1]

13 Jan 2020

Shorter birth intervals between siblings are associated with increased risk of parental divorce

PONE-D-19-28900R1

Dear Dr. Berg,

We are pleased to inform you that your manuscript has been judged scientifically suitable for publication and will be formally accepted for publication once it complies with all outstanding technical requirements.

With kind regards,

David Meyre

Academic Editor

PLOS ONE
---

## [Editor Report · Acceptance letter]

17 Jan 2020

PONE-D-19-28900R1 

Shorter birth intervals between siblings are associated with increased risk of parental divorce 

Dear Dr. Berg:

I am pleased to inform you that your manuscript has been deemed suitable for publication in PLOS ONE. Congratulations! Your manuscript is now with our production department. 

With kind regards,

on behalf of

Dr David Meyre 

Academic Editor

PLOS ONE